# Regulation of Survivin Isoform Expression by GLI Proteins in Ovarian Cancer

**DOI:** 10.3390/cells8020128

**Published:** 2019-02-06

**Authors:** Diana Trnski, Maja Gregorić, Sonja Levanat, Petar Ozretić, Nikolina Rinčić, Tajana Majić Vidaković, Držislav Kalafatić, Ivana Maurac, Slavko Orešković, Maja Sabol, Vesna Musani

**Affiliations:** 1Ruđer Bošković Institute, Bijenička cesta 54, 10000 Zagreb, Croatia; diana.trnski@irb.hr (D.T.); levanat@irb.hr (S.L.); pozretic@irb.hr (P.O.); nikolina.rincic@gmail.com (N.R.); tajana.majic-vidakovic@pporahovica.hr (T.M.V.); 2Zagreb Health School, Medvedgradska 55, 10000 Zagreb, Croatia; majagr79@gmail.com; 3PP Orahovica, Pustara 1, 33513 Zdenci, Croatia; 4Department of Obstetrics and Gynaecology, University Hospital Centre Zagreb, Petrova 13, 10000 Zagreb, Croatia; drzkal@gmail.com (D.K.); imaurac@yahoo.com (I.M.); oreskovic@hdgo.hr (S.O.); 5School of Medicine, University of Zagreb, Petrova 13, 10000 Zagreb, Croatia

**Keywords:** Hedgehog signaling, GLI proteins, survivin, ovarian cancer, isoform expression, polymorphisms

## Abstract

Ovarian cancer (OC) is the most lethal female gynecological malignancy, mostly due to diagnosis in late stages when treatment options are limited. Hedgehog-GLI (HH-GLI) signaling is a major developmental pathway involved in organogenesis and stem cell maintenance, and is activated in OC. One of its targets is survivin (*BIRC5*), an inhibitor of apoptosis protein (IAP) that plays a role in multiple processes, including proliferation and cell survival. We wanted to investigate the role of different GLI proteins in the regulation of survivin isoform expression (WT, 2α, 2B, 3B, and Δex3) in the SKOV-3 OC cell line. We demonstrated that survivin isoforms are downregulated in GLI1 and GLI2 knock-out cell lines, but not in the GLI3 knock-out. Treatment of GLI1 knock-out cells with GANT-61 shows an additional inhibitory effect on several isoforms. Additionally, we examined the expression of survivin isoforms in OC samples and the potential role of *BIRC5* polymorphisms in isoform expression. Clinical samples showed the same pattern of survivin isoform expression as in the cell line, and several *BIRC5* polymorphisms showed the correlation with isoform expression. Our results showed that survivin isoforms are regulated both by different GLI proteins and *BIRC5* polymorphisms in OC.

## 1. Introduction

Most malignant ovarian tumors in adult women are epithelial ovarian tumors. These neoplasms are classified into different morphological categories according to the appearance of epithelial cells in serous, mucinous, endometrial, clear cell carcinomas, mixed, squamous, transient, and non-differentiated tumors [1]. Serous adenocarcinomas are the most common and make up 75% of all ovarian cancer. Each histological type of ovarian tumor and the degree of differentiation of the cells is associated with certain molecular-genetic changes.

The Hedgehog signaling pathway is a developmental pathway involved in formation of various tissues and organs, including the ovary. In mammals, canonical signal transduction is triggered by binding of the ligand Hedgehog (HH) to the transmembrane receptor Patched (PTCH). This leads to receptor internalization and exposure of the protein Smoothened (SMO) on the cell surface. Activation of SMO triggers the release of transcription factors GLI (GLI1-3) from Suppressor of Fused (SUFU) and the translocation of GLI to the nucleus. GLI, then, activates transcription of HH target genes which are involved in cell cycle regulation, proliferation, adhesion, epithelial-mesenchymal transition, self-renewal, and pathway autoregulation [2,3]. Recently, survivin has also been identified as a novel HH target gene. Vlčková et al demonstrated that expression of survivin is directly regulated by GLI2, more specifically by the GLI2ΔN isoform [4]. A study by Brun et al showed that survivin is overexpressed in HH-driven medulloblastoma. They suggest it may represent a novel therapeutic target for this disease [5].

Survivin is the smallest member of the inhibitors of apoptosis (IAP) family of proteins in mammals [6]. The gene for survivin, *BIRC5* (baculoviral IAP repeat containing 5), is located on human chromosome 17q25 [7]. The *BIRC5* gene has four dominant exons (1, 2, 3, and 4) and two cryptic exons (2B and 3B). Alternative splicing of its pre-mRNA produces at least five different mRNAs encoding five different proteins with different intracellular distribution and anti-apoptotic properties: wild type survivin (S WT), survivin 2α (S 2α), survivin 2B (S 2B), survivin 3B (S 3B), and survivin-ΔEx3 (S ΔEx3) (Figure 1) [8].

S WT, discovered in 1997, contains a BIR domain necessary for its anti-apoptotic function [9]. It is expressed during development but is not expressed in most differentiated adult tissues. The overexpression of survivin is common in almost all tumors and is indicative of decreased overall survival, increased rate of recurrence, and resistance to therapy [6].

S 2α is the smallest isoform with a truncated BIR domain [10]. The reports about its function in cancer is conflicting, while some studies report that it binds to and attenuates anti-apoptotic activity of WT survivin or correlates with expression in lower stages of the disease [8,10], other studies report its association with resistance to treatment [11,12].

S 2B is the longest survivin isoform, but the insertion of cryptic exon 2B interrupts the BIR domain [13]. Its function remains unclear. While some studies report that it promotes cell death, or that its expression is inversely correlated with the tumor stage [8,11,14,15,16,17], others report its expression being correlated with treatment resistant cancers [18].

S 3B lacks the carboxyl-terminal coiled-coil domain of WT survivin [19]. It is considered cytoprotective [20,21] and its overexpression has been correlated with shorter overall survival and resistance to therapy [11,22].

S Δex3 contains a bipartite nuclear localization signal (NLS) and localizes to nucleus in cancer cells [23,24]. Its expression in tumors is associated with aggressive disease and unfavorable prognosis [24,25].

Survivin exhibits cell-cycle-dependent expression that is mostly controlled at transcription level. Its accumulation during mitosis is also influenced by posttranslational modifications that affect its stability. When expressed during mitosis, it is located in various sections of the mitotic apparatus such as centrosomes, microtubules, and anaphase spindles, and remains of the mitotic apparatus [6]. The homologous deletion of survivin results in early embryo death, which shows its crucial role in cellular development, differentiation, and homeostasis. It is selectively expressed in cancer cells, but not in healthy tissues. Excess accumulation of survivin is associated with the development of disease, disease recovery, and prognosis in various cancers, including bladder cancer, cervix, head and neck, prostate, skin, and ovarian [7]. A global deregulation of the *BIRC5* gene mediated by oncogenes (including STAT3, E2F or mutated RAS) or by the loss of tumor suppressors such as p53 or APC, accounts for the selective expression of survivin in cancer [26]. Since survivin is expressed exclusively in cancer cells, it is an interesting target for targeted therapy and new methods for detection of survivin, as well as new inhibitors, are constantly being developed [18,26,27,28].

Several *BIRC5* polymorphisms have been studied and have been associated with susceptibility to lung [29], gastric [30], bladder [31], oral [32], and liver cancer [33] as well as age of onset in ovarian cancer [7] and survival in colorectal cancer [8] and breast cancer [34].

X-ray crystallography has shown that survivin is organized as a dimer [6]. Since its isoforms exhibit various apoptosis-related properties, it is believed that the formation of survivin heterodimers with its isoforms may be important for regulating the function of survivin [8]. Since survivin isoforms can affect the activity of wild type survivin, and the transcription of this gene is regulated by HH-GLI signaling, it is important to investigate which factors contribute to the expression of certain isoforms. In this paper our goal was to examine the role of individual GLI transcription factors in the transcriptional regulation of survivin isoforms.

## 2. Materials and Methods

### 2.1. Generation of Knockout Cell Lines

SKOV-3 cell line [3] was maintained in DMEM supplemented with 10% FBS and penicillin/streptomycin. The CRISPR/Cas9 system was used to generate the knock-out cell lines. sgRNA molecules targeting the area surrounding the STG site of GLI1, GLI2, and GLI3 genes were generated using the online tool at crispr.mit.edu web site. Five top ranking sgRNA molecules were selected, and the one closest to the ATG site of each gene was selected. The sgRNA oligos were annealed, phosphorylated and cloned into the pX330-U6-Chimeric_BB-CBh-hSpCas9 (plasmid #42230, Addgene, Watertown, MA, USA) [35] into the BbsI restriction site, and the resulting vectors were sequenced on the ABI Prism 310 sequencing machine. The oligo sequences are listed in Appendix A. The SKOV-3 cell line was transfected with the generated vectors using the Xfect transfection reagent (Clontech, Mountain View, CA, USA). At 24 h post-transfection, the cells were trypsinized and plated into 96-well plates at the density of 1 cell/well. Several cell lines were propagated from single cells, and protein expression for each of the three GLI proteins was examined by Western blot to select the lines with the best knock-out for each protein.

### 2.2. Cell Culture Experiments

Cells were plated in 6-well plates, treated with two concentrations of GANT-61 (10 and 20 µM). The cells were collected 24 h after treatment. For the transfection of GLI1 (pcDNA4NLSMTGLI1, a kind gift from Prof. F. Aberger, Austria), GLI2 (p4TO6MTGLI2, a kind gift from Prof. M. Stevanovic, Serbia), GLI3 (pcDNA4/TO/GLI3richtig, a kind gift from Prof. M. Stevanovic, Serbia), and GLI3R (EGLI3-PHS, a kind gift from Prof. R. Toftgard, Sweden) the cells were seeded in 6-well plates and transfected the next day with 5 µg of plasmid DNA using the X-fect reagent (Clontech) following the manufacturer’s instructions.

### 2.3. Clinical Samples

Forty ovarian carcinoma (OC) samples and nine healthy fallopian tubes (FT) tissue samples (excised for reasons other than malignant transformations) were collected at the Department of Obstetrics and Gynaecology, University Hospital Centre Zagreb, School of Medicine, University of Zagreb. Patient blood samples were also collected. Blood samples collected from 74 healthy elderly women with no personal history of ovarian cancer were used as control for comparing polymorphism frequencies. 

All patients gave their informed consent before the samples were taken, and samples were collected with the approval of the hospital’s Ethical Committee (number of approval 02/21 AG, issued by the University Hospital Centre Zagreb on March 7th, 2017). The study was carried out following the rules of the Declaration of Helsinki Principles. All tissue samples taken during surgery were immediately placed in a vial containing 1 mL RNALater solution (Invitrogen, Carlsbad, CA, USA), kept at 4 °C overnight, and DNA and RNA were extracted the following day. DNA from tissue samples was extracted by the standard phenol-chloroform method and from blood by the salting out method.

### 2.4. Western Blot

Proteins were extracted by sonication in RIPA buffer supplemented with protease inhibitors (Roche, Basel, Switzerland). Protein concentration was determined using the Pierce BCA Protein Assay Kit (Thermo Fisher, Waltham, MA, USA). Western blot was performed as previously described [36] using the following primary antibodies: GLI1 (V812, Cell Signaling Technology, 1:1000, Danvers, MA, USA), GLI2 (ARP31885_T100, Aviva Systems Biology, 1:1000, San Diego, CA, USA), GLI3 (19949-1-AP, ProteinTech, 1:600, Rosemont, IL, USA), GFP for detection of GFP tagged GLI3R (Abcam ab6556, 1:1000, Cambridge, UK), and β-actin (60008-1-Ig, ProteinTech, 1:4000) and γ-tubulin (Santa Cruz Biotechnology, sc-7396, 1:500, Dallas, TX, USA) as loading controls.

### 2.5. Expression Analysis 

RNA was extracted using TRIZol reagent (Thermo Fisher) and reverse-transcribed using TaqMan Reverse Transcription Reagents (Applied Biosystems, Foster City, CA, USA). qRT-PCR was performed on a CFX96 machine (Bio-Rad Laboratories, Hercules, CA, USA) using Sso Fast EVAGreen Supermix (Bio-Rad).

The PCR conditions were as follows: initial denaturation at 95 °C for 3 min, 40 cycles of 95 °C for 15 sec, 61 °C for 1 min, and finally melting curve from 70 °C to 95 °C. All experiments were performed at least in duplicate. Primer sequences are listed in Appendix A. Expression levels were calculated using the 2^−ΔΔCt^ formula, relative to the housekeeping gene *TBP* [37]. For samples with no expression of a specific target after 40 cycles of qRT-PCR, the Ct was set to 40 to enable the statistical analysis of data [38].

### 2.6. Genotyping

Whole coding region (including alternate exons S 2α, S 2B, and S 3B) was genotyped, including the six SNPs in the *BIRC5* promoter and four SNPs in the 3′UTR region selected from the National Center for Biotechnology Information SNP database (http://www.ncbi.nlm.nih.gov/snp). Primers were designed using the Primer3 online tool (http://bioinfo.ut.ee/primer3/) [39]. Thirteen PCR fragments were analyzed using high resolution melting analysis on High-Resolution Melter (HR-1, Idaho Technology, Salt Lake City, UT, USA) as described in Cvok et al, 2008 [40] followed by Sanger sequencing (ABI PRISM 310 Genetic Analyzer, Applied Biosystems). Due to presence of four different polymorphisms in the PCR product of exon 4 and the beginning of 3′UTR, it was directly sequenced. PCR primer sequences and cycling conditions are listed in Appendix A. Due to high GC content in the DNA sequence of promoter region, in all PCR fragments located in the promoter CG RICH buffer (Roche) was added.

### 2.7. Statistical Analysis

D'Agostino–Pearson test was used for testing distribution normality of the expression data. For variables which showed normal distribution after logarithmic transformation, independent sample t-test and one-way analysis of variance were used to test the differences in isoforms’ expressions. Otherwise Mann–Whitney test and Kruskal–Wallis test were used. Nonparametric Spearman rank correlation coefficient (ρ) was used to assess the correlation of expression between various isoforms. For comparing polymorphism frequencies Fisher’s exact test (2 × 2) and χ^2^ test (3 × 2) were used. Two-tailed *p* values less than 0.05 were considered statistically significant. Statistical analysis was performed using MedCalc, version 18.2.1 (MedCalc Software bvba, Ostend, Belgium).

## 3. Results and Discussion

### 3.1. GLI Regulation of Survivin Isoform Expression

To test the contribution of each GLI protein to the expression levels of five different survivin isoforms, we generated SKOV-3 knock-out lines for GLI1, GLI2, and GLI3 proteins and examined the expression of these isoforms by qRT-PCR. All three KO cell lines exhibit reduced levels of full length GLI proteins, as well as GLI2 and GLI3 repressor forms. However, the band corresponding to the GLI2ΔN isoform is still present in the SKOV-3 GLI2 KO cell line (Appendix A). The expression levels of all isoforms were significantly downregulated in the SKOV-3 GLI1 KO cell line. In the SKOV-3 GLI2 KO cell line, only the S 3B and S Δex3 isoforms were downregulated, whereas knocking out GLI3 had no effect on either isoform (Figure 2). Therefore, we decided to focus on the SKOV-3 GLI1 KO line.

SKOV-3 wild type (WT) and the GLI1 KO lines were treated with two doses of the GLI1/2 inhibitor GANT-61 to test the responsiveness of different isoforms. GANT-61 showed a dose-dependent additive effect to knocking out GLI1 on the expression of S WT, S 2α, and S 2B, which suggests that these three isoforms are regulated by both GLI1 and an additional GANT-61 target, probably GLI2. The S 2B isoform shows the strongest response to inhibition with GANT-61, and even though it is affected by the GLI1 KO, it is less pronounced than in the other isoforms, suggesting that this isoform is regulated primarily by GLI2. The lack of effect on expression of these isoforms in the GLI2 KO cell line may be due to the persisting GLI2ΔN isoform, which is known to be a much more potent activator than the GLI2 full length protein [41]. The S 3B and S Δex3 isoforms do not respond to additional GANT-61 inhibition, suggesting that they are primarily regulated through GLI1 (Figure 3).

Our results correspond to those of Vlčkova et al [4], who showed that GLI1 and GLI2 have a moderate effect on survivin promotor activation and GLI3 displayed no promotor activation. In their study, they studied the GLI2ΔN isoform separately and showed that this GLI2 isoform has the strongest activation potential. 

To validate the role of specific GLI transcription factors in the expression of survivin isoforms we overexpressed all three GLI transcription factors in the SKOV-3 cell line, including the GLI3 repressor (GLI3R) (Appendix A). Transfection of GLI1 and GLI2 expectedly activated the HH-GLI pathway, detected by the upregulation of *GLI1* and *PTCH1* genes. Overexpression of GLI3 showed no effect on HH-GLI pathway activity, whereas overexpression of GLI3R downregulated the pathway (Figure 4A). Only slight changes in gene expression levels of survivin isoforms were observed upon GLI transcription factor transfection, probably due to an already high level of activity of these genes in cancer cells. On the other hand, GLI3R overexpression, which downregulates the HH-GLI signaling pathway (Figure 4A), leads to a significant decrease in expression of all survivin isoforms, except S 2B (Figure 4B).

### 3.2. Expression of Splice Variants

To verify that the SKOV-3 cell line is the appropriate model for studying survivin isoforms in OC, we compared basal expression levels of all five isoforms in cells to those in OC tissues. The expression patterns were shown to be the same (*p* < 0.001), implying that this cell line is a valid model for studying survivin isoforms in OC (Figure 5). 

The mRNA expression of S WT, S 2α, S 2B, S 3B, and S Δex3 was analyzed in 29 OC samples and compared to FT. Isoforms S WT and S 2α were detected in 96.5% of OC samples, S 2B in 58.6%, S 3B in 68.9%, and S Δex3 in 89.6% of OC samples. In contrast, the only splice variants detected in healthy FT samples were the weakly expressed S WT and S 2α. No other isoforms were detected in FT. The levels of all survivin isoforms expression differed significantly from each other (*p* < 0.001) (Table 1). The highest expression was determined for the S WT and S 2α isoforms, while S 2B had the lowest expression. OC samples showed significantly higher expression of these two isoforms compared to healthy fallopian tube tissue (*p* = 0.0001 and *p* = 0.0015, respectively) (Figure 6).

The expression levels of survivin isoforms was compared to the levels of GLI1, GLI2, and GLI3 expression which were published previously for the same set of OC tissue samples (data not shown) [36]. The only isoform which was correlated with the GLI genes was the S 2B isoform, which correlated with GLI2 expression (*ϱ* = 0.55, *p* = 0.019). Interestingly, in SKOV-3 cells the S 2B isoform showed significant upregulation only after transfection of GLI2 (Figure 4B), indicating that this isoform is regulated by GLI2.

### 3.3. Genotyping

Fifteen different polymorphisms were detected in the analyzed samples (OC and control) (Table 2). Most of the polymorphisms were located in the 5' and 3′UTR regions including the promoter region. All polymorphisms were in the Hardy–Weinberg equilibrium. There was no significant difference in distribution of genotypes and alleles between OC samples and controls for any of variant. However, c.-235A variant was more frequent in OC but with borderline significance (*p* = 0.053). Likewise, the frequency of heterozygous genotype c.-235G>A was higher in OC with marginal significance (*p* = 0.051) (Appendix A).

The allele frequencies for polymorphisms were in accordance with previous reports [29,34,42,43,44,45,46] or in NCBI database (http://www.ncbi.nlm.nih.gov).

### 3.4. Linkage Disequilibrium

Analysis showed there is a difference in number of variants in linkage disequilibrium between OC samples and controls. OC cases showed increased non-random association of alleles at multiple loci. In case of c.-1547C>T and c.9386T>C, these two polymorphisms always appear in the same combination in all tested OC samples showing complete linkage disequilibrium (r^2^ = 1.0) (Figure 7).

LD has often been reported between various *BIRC5* polymorphisms. Most of the correlations found here have already been described [29,34,44,47,48]. Our group recently described the LD between c.-644C>T and c.9809T>C in breast cancer (submitted). LD between c.235T>C and c.221+209T>C found in almost total LD, has not been described before.

### 3.5. Correlation of Splice Variant Expression with Genetic Polymorphisms in OC Patients

Seven polymorphisms showed significant association with expression of survivin isoforms. 

c.-1547C>T showed the highest association with splice variant expression (Appendix A). Samples with homozygous c.-1547T/T genotype showed significantly higher expression of isoforms S 2α (*p* = 0.016), S 2B (*p* = 0.036), and S 3B (*p* = 0.028). Major T allele was also significantly associated with higher expression of S 2α (*p* = 0.033), and S 3B (*p* = 0.045) than C allele.

c.9194G>A also showed significant association with variant expression (Appendix A). Samples with heterozygous GA genotype had significantly higher expression of S WT (*p* = 0.0001) and S 2α (*p* = 0.0005) than AA genotype. Minor G allele was also significantly associated with higher expression of S WT (*p* < 0.0001) and S 2α (*p* < 0.0001) compared to A allele.

c.9386T>C and c.10611C>A showed significant association with the expression of S 2α and S 3B (Appendix A). Samples with major c.9386TT genotype had significantly higher expression of S 2α (*p* = 0.032) and S 3B (*p* = 0.036) than TC genotype. Samples with major c.10611CC also had significantly higher expression of S 2α (*P =* 0.040) and S 3B (*p* = 0.036) than CA genotype.

c.-625G>C, c.-235G>A, and c.221+209T>C also showed some significant associations with isoform expression (Appendix A). Samples with minor homozygous c.-625CC genotype had significantly lower expression of S 2α in comparison to heterozygous GC genotype. Samples with major G allele of c.-235G>A (*p* = 0.047) or major T allele of c.221+209T>C (*p* = 0.045) had significantly higher expression of S 3B isoform than the minor alleles.

Association of *BIRC5* polymorphisms and expression of survivin isoforms have not been explored a lot in the literature. There are some reports about the influence of several polymorphisms on total survivin expression, most notably c.-31G>C [46,48,49,50], but also c.9386T>C [51] and c.9809T>C [47]. Antonacopoulou et al studied the correlation of c.-31G>C and c.9386C>T with the survivin isoform expression in colorectal cancer, and they found the same association of c.9386C>T genotypes with the expression of S 2α [8].

## 4. Conclusions

We have detected survivin isoform expression (S WT, S 2α, S 2B, S 3B, and S Δex3) in the ovarian cancer cell line SKOV-3, that follows the trend of isoform expression levels in its GLI KO variants and in ovarian cancer samples (Figure 5). The *BIRC5* gene has been recognized as one of the transcriptional targets of the HH-GLI signaling pathway [4]. The GLI2ΔN isoform has been indicated as the main transcription factor that regulates *BIRC5* gene expression. We have previously detected activity of the HH-GLI pathway in ovarian cancer [36] and therefore we wanted to examine the role of the GLI transcription factors not only in the regulation of survivin wild type expression, but also its isoforms. Because of the different roles of certain isoforms and their ability to regulate survivin function after heterodimer formation it is important to investigate how certain isoforms are regulated. Interestingly, our results confirm that survivin is regulated by the HH-GLI signaling pathway, but not all isoforms are regulated in the same manner. More specifically, all isoforms are regulated by GLI1, but isoforms S WT, S 2α, and S 2B can additionally be inhibited with the GLI1/2 inhibitor GANT-61, suggesting that these isoforms can also be regulated through other GANT-61 targets, most likely, GLI2. 

Seven *BIRC5* polymorphisms are also associated with the expression of survivin isoforms. Therefore, there is an intricate regulation of expression between genetic influence and transcription factors such as GLI proteins to determine the final expression of survivin isoform expression and their anti- and pro-apoptotic role that could determine the fate of the cells.

## Figures and Tables

**Figure 1 cells-08-00128-f001:**
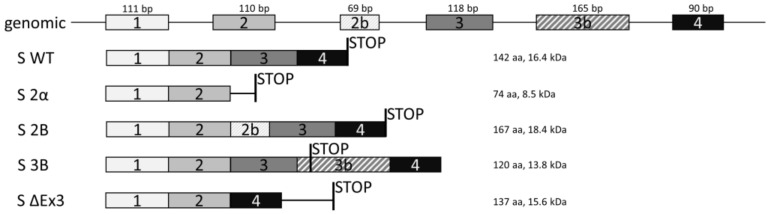
The exon structure of five splice isoforms of survivin. Exons are drawn relative to their size. Vertical bars indicate the site of stop codon of each isoform.

**Figure 2 cells-08-00128-f002:**
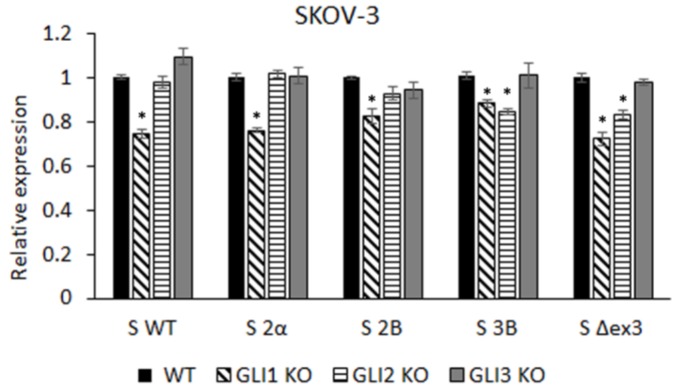
Levels of survivin isoform expression in the SKOV-3 GLI KO lines compared with SKOV-3 WT cells. * represents statistically significant downregulation in comparison with the WT cell line (*p* < 0.05).

**Figure 3 cells-08-00128-f003:**
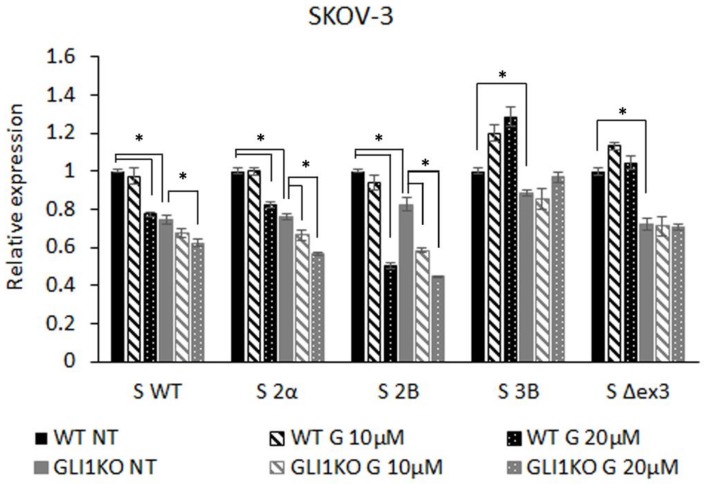
Survivin isoform expression levels in the SKOV-3 WT cell line and the SKOV-3 GLI1 KO cell line after treatment with the GLI1/2 inhibitor GANT-61 for 24h. NT, non-treated samples; G 10µM, cells treated with 10 µM GANT-61; G 20µM, cells treated with 20 µM GANT-61. * represents statistically significant downregulation (*p* < 0.05).

**Figure 4 cells-08-00128-f004:**
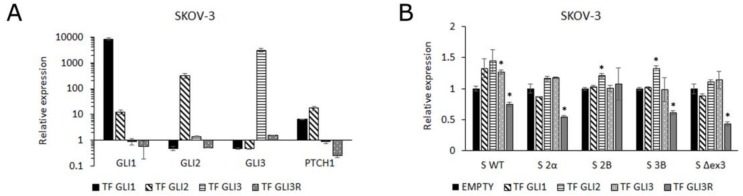
The effect of GLI1, GLI2, GLI3, and GLI3R overexpression in the SKOV-3 WT cell line on (**A**) the expression levels of *GLI1*, *GLI2*, *GLI3*, and *PTCH1* and (**B**) the expression levels of survivin isoforms. * represents statistically significant changes in expression compared with cells transfected with empty vector.

**Figure 5 cells-08-00128-f005:**
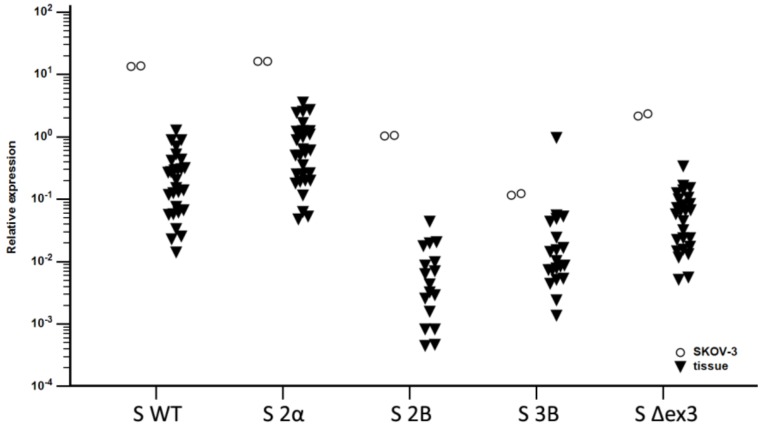
Relative expression of survivin isoforms in the SKOV-3 cell line (circles) and ovarian carcinoma (OC) tissue samples (triangles).

**Figure 6 cells-08-00128-f006:**
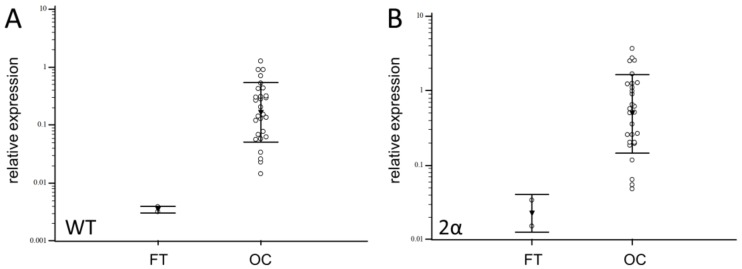
Relative expression of S WT (**A**) and S 2α (**B**) in fallopian tube (FT) and ovarian carcinoma (OC) samples.

**Figure 7 cells-08-00128-f007:**
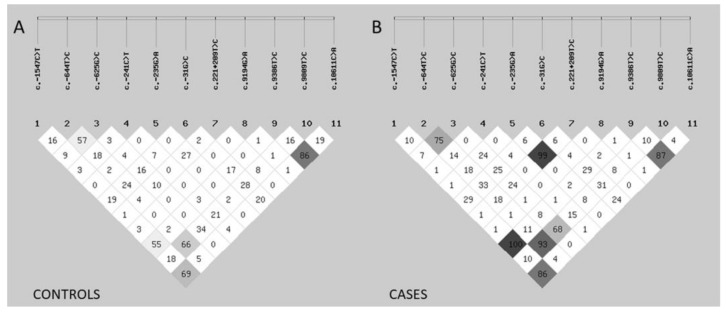
Pairwise linkage disequilibrium (LD) of eleven *BIRC5* sequence variants with highest minor allele frequency. The location of each sequence variant along the *BIRC5* gene is relative to the real nucleotide position. The number in each diamond indicates the intensity of LD (r^2^ × 10^2^) between respective pairs of sequence variants. The LD strength is also represented by shades of gray (0 [white] < r^2^ < 1 [black]). (**A**) Controls. (**B**) Cases.

**Table 1 cells-08-00128-t001:** The association of survivin isoform expression in OC samples.

S 2α	S 2B	S 3B	S Δex3	-	-
0.92	0.69	0.81	0.75	**S WT**	*ϱ*
<0.0001	0.002	<0.0001	<0.0001		*p*
-	0.67	0.80	0.81	**S 2α**	*ϱ*
-	0.005	<0.0001	<0.0001		*p*
-	-	0.55	0.67	**S 2B**	*ϱ*
-	-	0.021	0.004		*p*
-	-	-	0.66	**S 3B**	*ϱ*
-	-	-	0.002		*p*
-	-	-	-	-	-
Spearman rank correlation coefficient (*ϱ*)					

**Table 2 cells-08-00128-t002:** *BIRC5* polymorphisms in OC samples and healthy controls. MAF, minor allele frequency; OC, ovarian carcinoma.

Gene Region	SNP ID Number	Nucleotide Change	MAF in OC (%)	MAF in Controls (%)	*p*-Value
promoter	rs3764383	c.-1547C>T *	23.1	25.0	0.871
promoter	rs143396310	c.-1458C>T	1.3	1.4	1.000
promoter	rs8073903	c.-644T>C	26.3	33.1	0.297
promoter	rs8073069	c.-625G>C	21.3	22.3	1.000
promoter	rs17878731	c.-267G>A	1.3	0.7	1.000
promoter	rs17878467	c.-241C>T	5.0	10.8	0.219
promoter	rs17887126	c.-235G>A	6.3	1.4	0.053
5′UTR	rs9904341	c.-31G>C	48.8	37.2	0.093
intron 2	rs4789551	c.221+209T>C	6.4	4.7	0.756
exon 4	rs2071214	c.9194G>A **	3.8	3.4	1.000
3′UTR	rs17885521	c.9288G>C	1.3	2.0	1.000
3′UTR	rs17882627	c.9342G>A	2.5	1.4	0.614
3′UTR	rs2239680	c.9386T>C	23.8	23.0	1.000
3′UTR	rs1042489	c.9809T>C	25.0	35.8	0.104
3′UTR	rs2661694	c.10611C>A	26.3	25.7	1.000

* C is the minor allele, ** G is the minor allele.

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
