# Peer review of "Regulation of Survivin Isoform Expression by GLI Proteins in Ovarian Cancer"

_cells, 2019, doi:10.3390/cells8020128_

Round 1

Reviewer 1 Report

Trnski et al., investigate in their manuscript in details a correlation between GLI and various forms of survivin. Interestingly in this manuscript, a correlation between genetic polymorphism of surviving and its splice variants were analyzed which is a very rare thus intriguing study. Overall, this manuscript presents good pieces of evidence for an effect of GLI proteins on survivin and gives the reader a lot of information about the correlation between those. This manuscript is prepared in a soundly way and data are analyzed with proper statistical tests showing significant differences.

Minor concerns:

1.     Indicate statistical significance (with an * and description in the Figure legend) for Figure 2 and 3.

2.     Use “point” instead of “comma” to indicate decimals (see Y-axis in Figure 2, 3, 4).

3.     Prepare all graphs in more uniform way e.g. do not use helper horizontal lines in the graphs, use the same style of axis, the same thickness of lines, the same style of fonts in the graphs etc.

Author Response

Thank you for your kind comments.

1.     Indicate statistical significance (with an * and description in the Figure legend) for Figure 2 and 3.

2.     Use “point” instead of “comma” to indicate decimals (see Y-axis in Figure 2, 3, 4).

Response: We redrew the figures, added the asterisk where necessary and corrected the decimal points.

3.     Prepare all graphs in more uniform way e.g. do not use helper horizontal lines in the graphs, use the same style of axis, the same thickness of lines, the same style of fonts in the graphs etc.

Response:  We also tried to make all the graphs more uniform.

Reviewer 2 Report

This paper reports on a study of the regulation of expression of survivin isoforms such as WT,2α, 2B, 3B, ex3 by GLI1, GLI2 and GLI3 proteins. The Authors have generated SKOV-3 knock-out (KO) lines for these proteins and have checked expression of survivin using qRT-PCR technique. First, the expression of GLI proteins was investigated in SKOV-3 and SKOV-3 KO lines by Western blot. It was shown, that the level of GLI proteins was reduced in KO cells. The Authors have found that survivin isoforms were downregulated in GLI1 and GLI2 KO cell lines. They have also investigated the influence of GLI1 and two kinds of inhibitors - GANT-61 and GLI transcription factors including the GLI3 repressor on survivin isoforms expression. The comparison of mRNA expression of survivin isoforms in ovarian carcinoma (OC) and healthy fallopian tubes (FT) samples was also shown.  The Authors were also successful in BIRC5 polymorphism detection in ovarian carcinoma and control samples.

I recommend the manuscript for publication after minor revision according to the remarks presented below.

1.                  Abbreviations in Legends of the graphs should be more informative and significant. For example SKOV G10 is unclear for Readers. It should be explain that it is related to 10 µM concentration of GANT-61.

2.                   Figure Captions should be improved and list more details concerning the experimental conditions, such as solutions, pH, salt concentration, temperature, etc.

3.                  In Figure 2, 3, and 4B, commas in numerical values should be replaced with dots.

4.                  Figure 1:  bands for the third and fourth isoforms, and also those for the fourth and fifth isoforms are clearly not proportional to their molar masses, so do those for other isoforms as well. A note in Figure caption should be added informing that band lengths do not represent the molar mass and they only reflect qualitatively the main aspects of the composition, or that they not drawn to scale. Otherwise, the lengths of each element in the bands should be adjusted to reflect the actual portion of the respective genomic element.

5.                  Recently, survivin mRNA and protein have been analyzed in cancer cells using molecular beacon probes and antibody, see for instance: ACS Appl. Mater. Interfaces 2018, 10, 17028−17039; Biosensors and Bioelectronics 2016, 84, 37–43; Nanomaterials 2018, 8, 510; doi:10.3390/nano8070510. These relevant literature references should be cited.

6.                  Words such as Figure, Table, should be in uppercase font.

Author Response

Thank you for your kind comments and suggestions.

1. Abbreviations in Legends of the graphs should be more informative and significant. For example SKOV G10 is unclear for Readers. It should be explain that it is related to 10 µM concentration of GANT-61.

2. Figure Captions should be improved and list more details concerning the experimental conditions, such as solutions, pH, salt concentration, temperature, etc.

Response: We made the Figure legends more informative, and explained the abbreviations.

3.                  In Figure 2, 3, and 4B, commas in numerical values should be replaced with dots.

Response: We corrected the decimal points in the graphs.

4.                  Figure 1:  bands for the third and fourth isoforms, and also those for the fourth and fifth isoforms are clearly not proportional to their molar masses, so do those for other isoforms as well. A note in Figure caption should be added informing that band lengths do not represent the molar mass and they only reflect qualitatively the main aspects of the composition, or that they not drawn to scale. Otherwise, the lengths of each element in the bands should be adjusted to reflect the actual portion of the respective genomic element.

Response: We redrew the Figure 1 more to the scale of the relative exon sizes and added the Stop codon markings for clarity.

5.                  Recently, survivin mRNA and protein have been analyzed in cancer cells using molecular beacon probes and antibody, see for instance: ACS Appl. Mater. Interfaces 2018, 10, 17028−17039; Biosensors and Bioelectronics 2016, 84, 37–43; Nanomaterials 2018, 8, 510; doi:10.3390/nano8070510. These relevant literature references should be cited.

 Response: We added two of the references you suggested, one for the protein and one for mRNA detection.

Thank you for pointing us to these very recent publications. We capitalized the words Figure and Table as suggested.